# Digitized ADOS: Social Interactions beyond the Limits of the Naked Eye

**DOI:** 10.3390/jpm10040159

**Published:** 2020-10-08

**Authors:** Harshit Bokadia, Richa Rai, Elizabeth Barbara Torres

**Affiliations:** 1Department of Psychology, Rutgers University Center for Cognitive Science, The State University of New Jersey, Rutgers, NJ 08854, USA; hb271@scarletmail.rutgers.edu (H.B.); rr708@psych.rutgers.edu (R.R.); 2Department of Computer Science, Computational Biomedicine Imaging and Modelling Centre, Rutgers, The State University of New Jersey, Rutgers, NJ 08854, USA

**Keywords:** digital biomarkers, wearables, time series analysis, autism, social dyads, socio-motor parameters, network connectivity, non-linear complex dynamics, stochastic analysis

## Abstract

The complexity and non-linear dynamics of socio-motor phenomena underlying social interactions are often missed by observation methods that attempt to capture, describe, and rate the exchange in real time. Unknowingly to the rater, socio-motor behaviors of a dyad exert mutual influence over each other through subliminal mirroring and shared cohesiveness that escape the naked eye. Implicit in these ratings nonetheless is the assumption that the other participant of the social dyad has an identical nervous system as that of the interlocutor, and that sensory-motor information is processed similarly by both agents’ brains. What happens when this is not the case? We here use the Autism Diagnostic Observation Schedule (ADOS) to formally study social dyadic interactions, at the macro- and micro-level of behaviors, by combining observation with digital data from wearables. We find that integrating subjective and objective data reveals fundamentally new ways to improve standard clinical tools, even to differentiate females from males using the digital version of the test. More generally, this work offers a way to turn a traditional, gold-standard clinical instrument into an objective outcome measure of human social behaviors and treatment effectiveness.

## 1. Introduction

The wearable sensors revolution has brought behavioral science to a new era of precision. Across many research areas in basic and translational sciences, it is now possible to monitor natural motions continuously, as they unfold between two people in a social dyad, or even as part of a social group. Indeed, the advent of new advances in digital technology opens many new avenues of inquiry in the social, clinical, and psychological sciences. It is now possible to track multi-layered activities generated by our nervous systems while our brain controls our bodies in motion. Such tracking affords various levels of discourse, including a macro- and a micro-level of description.

At a macro-level (Figure 1A), we can describe the overt motions of our body to some degree, i.e., the motions that we can unambiguously perceive and explicitly describe. However, it is at the micro-level of analysis where information that transpires largely beneath awareness can help us understand the patterns of socio-motor behaviors underlying social interactions (Figure 1B). From facial micro-expressions to voice micro-fluctuations to bodily micro-gestures, all these non-verbal forms of communication permeate everything that we share socially with others, ever since birth (and even in uterus as we interact with our mother’s biorhythms [1,2]) in pre-cognitive stages of neurodevelopment. We are born with the capacity for social readiness and our perceptual systems help us make sense of the social world, whenever we receive appropriate support from an early age. Such support comes from our parents and from other social beings whose speech cadence and facial and bodily patterns we spontaneously imitate from an early age [3,4,5] through synchronous patterns and socio-motor entrainment.

There is a form of autonomous mirroring that tends to occur in social interactions from birth, when within a few weeks human babies can entrain their bodily biorhythms to the rhythmic speech of their mother [6,7] and perceive changes in facial micro-gestures with emotional content [8,9,10,11]. From birth, babies can distinguish biological motions [12,13], and by eight months of age they can dissociate structural and dynamic information from non-biological vs. biological motions [14]. Between four and five years of age, their socio-motor kinematics, to communicate decisions and desires through pointing gestures, mature into fundamentally different statistical signatures [15,16]. Their statistics transition from random and noisy to predictive signals necessary for proper motor control in order to start schooling and formal instruction [17,18].

The body of work on human development and the capacity of human babies for social readiness has relied on high grade instruments, microfilm, wearables, and other means beyond human observation. This is so because several of these pre-cognitive socio-motor behaviors are much too complex, variable and subtle to be captured by the naked eye [19,20]. They require sensors with proper sampling resolutions, to register the micro-fluctuations in biorhythmic activities across the nervous systems of the two people engaged in the social exchange. Using such means, it is then possible to discern important aspects of the interaction and distinguish synchronous intent, mutual trust, joint goals and overall coordination that leads to good rapport, successful gestural communication and anticipatory control [21].

We know that people automatically sway together when they converse and when they joke [22,23], that their bodies entrain when they build social rapport and that they show disjointed patterns when communication fails [24]. When one of the parties prevails in excess over the other, and that person leads the interaction most of the time, there is no opportunity for the other party to communicate [24]. They then dance to a different beat [25]. What is supposed to be a social dyadic dialogue breaks into a dyadic monologue. One of the agents no longer participates as it should in a social exchange. How could we study such phenomena more systematically and help current research in autism, the disorder of the nervous system that leads to fundamental differences in socio-motor control [26] and social communication [27]?

One possible way to probe the social dyad in a systematic manner is by using as the backdrop of our experiment well-established and broadly adopted tests of social interaction that have already been standardized in some form (Figure 1C). One such a test is the gold standard to detect autism in clinical and in research settings: The Autism Diagnostic Observation Schedule (ADOS) [28]. This test was designed precisely to detect the extent to which a person fails to reach a set of criteria, rendering the interaction socially appropriate. The rater of the test provides a numeric score reflecting the severity of the inability to carry on an appropriate social exchange. The test is one sided in that the certified rater both poses social presses to evoke social overtures in response and actively participates in the exchange. Then, from the outcome of such interactions, the rater scores the responses of the other person. This is very taxing on the rater who is both interacting with and observing the person being scored. It is also taxing on the person being scored, because the test takes a long time (approximately one hour), while the information obtained is very discrete, necessarily missing key aspects of the exchange that escape the busy rater’s eyes. 

Social interactions naturally occur in a very different manner (i.e., no one is rating the other person as the interaction unfolds.) Natural interactions are extremely complex, dynamic, and flow organically in (at times) unpredictable manners. One of the advantages of using the ADOS as a backdrop of our study is precisely that it is conceived to allow for such fluidity using several tasks that can be administered in the order in which the situation affords, rather than in a fixed order. Yet, despite this fluidity, because of the standard ways in which each task is conceived, the overall test preserves a systematic manner of administration across different ages and different verbal capacities, ranging from minimally to fluidly verbal cases. 

The ADOS test has not officially included neurotypical controls to build a normative scale [29]. It is a criterion-based rather than a norm-based test, yet this does not preclude us from using this test to probe neurotypicals while applying our micro-level inquiry, paired with wearables. Combining wearables which co-register simultaneously the bodily motions of both the rater and the participant with high sampling frequency and extract information both at the macro- and micro-levels of the social exchange, can give us a very rich picture of the highly non-linear complex dynamics taking place during the interactions. We hypothesize that the digital ADOS will reveal new aspects of the shared space spanned by the cohesive dyadic micro-movements of the child and clinician.

Integrating macro-and micro-levels of analysis will augment in unprecedented ways our knowledge about action execution and action observation in this closed loop of highly variable, transient, and sustained dyadic cohesiveness (e.g., Figure 1C–E). We may provide biometrics of a social dialogue through bodily activity. informing us not only about the person’s capacity for social readiness, but more importantly instructing us on how-to best nurture that potential. We may even be able to broaden the potential to help the person’s socio-motor tendencies to adapt and converge toward the desirable social norms of any given culture. Furthermore, because such bodily activities are sampled with high frequency, and because the micro-fluctuations derived from biorhythms shift with context and pick up subtle nuances otherwise hidden to the naked eye of the observer, we will be able to add significantly to the information that the macro-level ADOS test already provides. It may be also possible to detect differences between females and males in the cohort when examining the dyadic interactions, as social rapport emerges between the child and clinician, and does so differently for males than for females across the ADOS tasks. More generally, we may be able to transition such valuable clinical tests from a quasi-static discrete method of detection, to a highly dynamic, continuous tracking tool, one capable of providing an outcome measure for any treatment’s effectiveness during the naturalistic activities of daily living. 

In the social sciences in general, this approach could advance more than one line of inquiry into the non-linear, dynamic, and stochastic complexity of micro-level social exchange, making these important aspects of our social being no longer hidden to the naked eye.

## 2. Materials and Methods

### 2.1. Ethics Information

This study was approved by the Rutgers University Institutional Review Board. The study is compliant with the Helsinki Act.

### 2.2. Design

This study used the tasks from the research grade ADOS-2 [28], a standard clinical test to evoke social situations, rate the child’s responses and detect autism in cases whereby social affect and repetitive restrictive behaviors impede proper social interactions with the rater. (The term Repetitive Restrictive Behaviors (RRB) is used here to be congruent with the clinical jargon of the ADOS test. According to our quantitative metrics, these motions reflect, rather, adaptation of the nervous systems in order to enhance kinesthetic reafferent feedback (i.e., feedback from intentional movements) which in autism shows highly random noisy patterns in relation to neurotypical controls [15]). 

The experiment consisted of four visits. In the first visit, the neurotypical control participants were administered the tasks from the ADOS-2 module that were most appropriate in terms of age and verbal capacity. The first visit also examined autistic subjects under similar ADOS-2 module criteria, i.e., using the most appropriate module according to the child’s age, verbal status, and cognitive capabilities (as determined by the rater clinician.) The autistic participants were familiar with one of the two raters but not the other. Controls only participated in visit one.

The second visit employed an ADOS-2 module that was compatible with the child’s verbal ability, even if it was easier than the most appropriate module used in visit one. The idea was to vary the module while keeping the rater familiar to the children in both visits. The third visit switched to a new, unfamiliar rater but kept the most appropriate module. The fourth visit kept that rater but changed the module to an easier one, feasible but not the most appropriate (if one were to use the ADOS-2 to detect autism.) In our case, we used this test as an experimental assay to evoke dyadic social interactions. Appendix A shows the tasks’ names (and initials used in Figures labels) along with the corresponding module numbers. The raters are certified clinicians naïve to the purpose of the study.

### 2.3. Sampling Plan and Inclusion/Exclusion Criteria

We tested a total of 29 participants, 10 females and 19 males (see Appendix A). There were 11 control children of school age (mean age 9.9 years old) and 18 autistic children (mean age 8.3 years old). In this set 26/29 dyads (10 females and 16 males) had completed the digitized ADOS with six synchronous sensors, three on the child and three on the clinician co-registering their motions in tandem) and these were included in visit one. The 3/29 dyads not included in these analyses had grids with 12 sensors, and/or the child had a diagnosis of Fragile X. These will be the focus of a different publication, so we include only idiopathic autism in the present work. Furthermore, eight autistic children returned in visits one and two, six returned to the lab in visits one–three and one in visits one–four. The details of these participants, along with the module delivered and ADOS-2 score assigned across all the visits, are summarized in Appendix A. Autistic children were referred to our laboratory by local clinics and by the community, with help from the New Jersey Autism Center of Excellence (https://njace.us/), funded by the New Jersey Governor’s Council for the Medical Research and Treatments of Autism.

Because of the data type and methods that we introduce for personalized behavioral analyses, this cohort was sufficient in size. We test the proposition that combining pencil-and-paper methods with digital means would reveal information about dyadic behaviors that we would miss using observation alone. 

We built indexes that combined the rater’s scores derived from observation with digital data collected using light wearable sensors. The sensors (APDM Opals, Portland, OR, USA) synchronously co-registered motions continuously at 128 Hz, thus providing big data for personalized analyses. We study here linear acceleration (for results using angular velocity, see Appendix A) co-registered synchronously across six sensors (Figure 1B.) The six sensors were positioned at the wrist and torso of the participants, conceptualized as an interconnected network. Sessions took place in a socially controlled environment, using the standard ADOS tasks and set up (Figure 1C). Tasks were administered by two clinicians, who were naïve to the experimental goals of the study. Both clinicians had ADOS certification for research reliability. 

### 2.4. Analysis Plan

We developed new data types and analytics to simultaneously track the participating agents of the dyad within the context of the social interactions that the ADOS-2 test evokes. Importantly, our analytical methods consider the biorhythmic activities of each participant (individually) and of the shared spatio-temporal parameters of the dyad acting in concert (Figure 1A). We co-registered synchronously the upper body motions of both participants using a grid of unobtrusive wearable biosensors (Figure 1B) and examined each sensor’s activity pairwise in relation to the other sensor’s activity. The moment by moment fluctuations in the amplitude of the signals from each sensor are normalized to account for anatomical differences that impact motion parameters. A new standardized data type coined micro-movement spikes (MMS) is then obtained together with various indexes derived from the frequency and temporal domains of these data.

An index of Dyadic Strength is built as a ratio whereby the numerator is derived from the Dyadic Coherence, using analyses in the frequency domain (Figure 2A,B). The denominator uses the Dyadic Variability extracted from the MMS, providing information about the fluctuations in signal amplitude in the time domain. In both Dyadic Coherence and Dyadic Variability, we sum the dyadic-interaction entries of the 6×6 matrices that we build from pairwise sensors’ computations. Figure 2C shows the parameterization of the MMS cross-coherence, phase, and frequency matrices. These were conceptualized as adjacency matrices and combined to represent dynamically evolving weight-directed graphs. We then adapted network connectivity analyses using the Brain Connectivity Toolbox of MATLAB [30,31] to uncover various relevant connectivity metrics across the six nodes. The shared activity in the dyadic interaction entries were then used to obtain the Dyadic Coherence quantity per unit time. As with Dyadic Variability, we obtained this for each task of the session in the order in which the tasks were administered. The Dyad Strength metric was thus built from the ratio of Dyad Coherence divided by Dyad Variability and normalized between 0 and 1.

We analyze data in the time and frequency domains and construct various socio-motor indexes. In the time domain, we examine the variability in MMS using in-home developed analyses of time series data. These include the characterization of the MMS as deviations from an empirically estimated mean, determined using maximum likelihood estimation (MLE) of the best continuous family of probability distributions fitting the data. Previous studies involving natural behaviors had revealed the continuous Gamma family as a good characterization of human digital data registered with these wearables [15,32,33]. The MMS trains were windowed according to an optimal time block yielding tight confidence intervals for the optimal parameters in an MLE sense. They were analyzed task by task, using the simultaneously co-registered data from the six sensors that the child and rater wore (Figure 1C). To that end, using the Earth Mover’s Distance (EMD) metric [34], we examine the pairwise differences in frequency histograms derived from the MMS of the linear acceleration (see angular speed in Appendix A) and build a 6×6 matrix whereby entries provide information about variability in the micro-fluctuations of linear acceleration amplitude for dyadic- and for self-interactions (Figure 1D). The dyadic-interaction variability was obtained by summing all dyadic entries of one of the circled blocks of this symmetric matrix. This quantity was then used as the denominator of a ratio coined Dyadic Strength.

We derived several adjacency matrices to represent these activities using dynamically evolving weighted directed graphs. Furthermore, we adapted various network connectivity analyses from brain research and used them here to characterize a grid of sensors registering the continuous naturalistic social interactions of a dyad. We focus on the cohesiveness of the dyadic interactions from the spatio-temporal spaces shared. We also examine the kinematic synergies of self-interactions in each dyad participant.

Among the connectivity metrics that we used, we specifically assessed possible differences between males and females across the cohort. We used the clustering coefficient metric, a measure of the degree to which nodes in a graph tend to cluster together. We used the Betweenness Centrality metric, measuring the extent to which a vertex lies on paths between other vertices. Vertices with high betweenness may have considerable influence within a network by virtue of their control over information passing between other vertices. Related to this, we examined the characteristic path length, the average shortest distance path (geodesic), denoting he number of edges in the shortest paths between all vertex pairs. In all cases we gathered the data generated by the dyad across the tasks and used the same number of data points in males and in females, to ask if, given a scatter of similar size, the values distributed significantly differ. Since we had fewer females (10) than males (16), we relied on the minimum number of values in the females and cycled through all the values of the males (bootstrapping from the data) to ascertain the degree to which they differed when the number of elements were identical for each parameter and for each cohort. To compare the groups, we used the rank-sum test and visualized the histograms and the scatters across all connectivity parameters scaled to range between 0 and 1. This normalization allowed us to visually compare males and females across network connectivity metrics and stochastic parameters.

### 2.5. Data Types

We used Euclidean norm (Equation (1)) to obtain the scalar value of acceleration from the tri-axial acceleration time series, sampled ${\vec{A}}_{c}\left(t\right)=\left(A_{c}{\left(t\right)}_{x},\;A_{c}{\left(t\right)}_{y},\;A_{c}{\left(t\right)}_{z}\right)$ at the sampling frequency of 128 Hz.
(1)$$A_{c}(t)=\sqrt{A_{c}{\left(t\right)}_{x}^{2}+A_{c}{\left(t\right)}_{y}^{2}+A_{c}{\left(t\right)}_{z}^{2}}$$

The resultant time series of peaks and valleys for an experiment’s participant (EP01), during the ADOS “Construction task”, and the rater are shown in Figure 1C. The sensors co-registered orientation data through a tri-axial gyroscope. Similarly, we computed the angular speed capturing rotational data (results shown in Appendix A).

These time series of linear acceleration and angular speed have peaks and valleys that fluctuate in amplitude and inter-peak interval times. We focused in this paper on the fluctuations in signal amplitude, taken as relative deviations from the empirically estimated Gamma mean. To estimate the mean, we fit the continuous Gamma family of probability distributions to the frequency histogram of the raw data peaks. The Gamma family fits the data in an MLE sense with 95% confidence intervals for the shape and the scale parameters, respectively [15]. Using these parameters, we estimated the Gamma moments and used the mean to obtain the absolute deviation of each point in the time series. We then construct our waveform of interest denoting micro-fluctuations in bodily biorhythms during social interactions, derived from the original raw data of linear acceleration (and angular speed). 

The peak amplitude data mean, shifted and centered at the empirically estimated mean, was then scaled to the real valued interval (0,1) using Equation (2)
(2)$$NormPeak=\frac{Peak}{Peak+{Avrg}_{\min\;\mathrm{to}\;\min}}$$

These unitless waveforms are called micro-movement spikes (MMS) (see Figure 1C). The MMS are in reduced frames as they capture only the absolute deviations from the estimated mean. We zero-pad values below the deviations to reconstruct the original number of frames across all six sensors. This standardized waveform accounts for allometric effects [35] owing to anatomical differences impacting the motion data and allows us to combine data from multiple sensors with different physical units [19]. Here we focus on the linear acceleration and present the results in the main paper. The Appendix A presents the results for angular speed.

We chose 12 s time windows as our unit for the Gamma parameter estimation because this was the minimal task segment across the entire set and it provided tight confidence intervals at 128 Hz sampling rate.

### 2.6. Socio-Motor Metrics

Several metrics per unit time (12 s windows independently sampled without overlap) are derived to quantify socio-motor phenomena, inclusive of leading patterns within the dyad agents, cohesiveness and phase delays. Below we enumerate these metrics in the temporal and frequency domains. Furthermore, we explain our adaptation of network connectivity analyses commonly used in brain science [30,31] to the analyses of interacting nodes on a grid of synchronous sensors, sampling biorhythmic activities generated by bodies producing social motions and emotions.

*Dyad Variability (temporal domain):* To quantify the differences in frequency histograms derived from the MMS, we use the Earth Mover’s Distance metric [36] commonly employed in machine learning [34]. The average pairwise EMD per unit time is taken across all time windows of each task, owing to variable task time lengths. We sample windows under the independent identically distributed (IID) assumption for each task and average across these uniform units of time. Thus, our results are reported as quantity per unit time, using our uniformly determined unit of time. For each child-clinician dyad, this information produces a 6×6 matrix with the first three rows representing the child data from the right wrist (R), left wrist (L) and the torso (T) sensors in that order. The following rows represent the clinician’s sensors in the same order as the child’s sensors. Likewise, the columns follow this convention such that entry [1,2] in row 1 and column 2 represents the pairwise EMD_1,2_ quantity between the right wrist and the left wrist of the child, whereas entry in row 5 and column 2 represent the EMD_5,2_ between the left wrist of the clinician and the left wrist of the child. EMD_1,2_ captures variability in bodily nodes’ self-interaction because these are from the child’s own body. In contrast, EMD_5,2_ captures variability in dyadic interactions because they are from a shared dyadic spatio-temporal pattern derived from pairwise activities between the clinician and the child. Figure 1D shows sample frequency histograms from the child and clinician right wrist MMS used to obtain the entry marked by an asterisk on the EMD matrix. The sum over the Child-Clinician (or the Clinician-Child) entries reflect Dyadic Variability (note this is a symmetric positive matrix as EMD is a proper distance metric.) This quantity is obtained for each task across the time length that the task may take, using 12 s long time windows under IID assumption. The average for each task is then plotted across the tasks (normalized by maximal value to plot it as (0,1) values). Figure 1E shows an example of this profile for EP01-clinician dyad.

*Coherence-Phase-Frequency Parameterization (frequency domain):* Power spectrum analyses (Figure 2A) of the MMS and pairwise cross-coherence analyses across the grid of sensors were used to build a parameterization of the data in the frequency domain. To that end, for each task we obtain the pairwise cross-coherence and cross-spectrum phase across frequencies, thus providing maximal cross-coherence with the corresponding phase and frequency, forming three matrices lined up in Figure 2C. These matrices are used to derive several socio-motor parameters using network connectivity analyses (see below).

The first matrix comprises the maximal cross-coherence value at each entry. For example, entry MaxCoherence_2,3_ represents the maximal cross-coherence value between the left wrist (L) of the child and the torso (T) of the child. This is the maximal level of synchronicity between these self-body parts. As with the EMD matrix before, here we focus on the self-interaction entries and on the dyadic entries, separately. Under this convention, entry MaxCoherence_2,3_ represents a self-interaction entry. The corresponding entry in the MaxPhase matrix has 0 value, thus indicating that these two self-body parts are synchronized with 0 lag and this occurs at 30Hz, the corresponding entry in the MaxFrequency matrix.

The MaxCoherence is conceptualized as an adjacency matrix representing a weighted connected graph. We use it to derive an interconnected network of nodes (body parts) and links between nodes. The links represent the level of pairwise maximal coherence. We use network connectivity analyses and obtain OutDegree, InDegree, total degree, strength, modularity, clustering coefficient, etc., to capture different states of network connectivity across tasks representing states of the dyad whose biorhythmic activity dynamically evolves over time. For the corresponding MaxPhase matrix, we adopt the convention that node *i* leads node *j* in the (*i*, *j*) entry. Lastly, the frequency matrix provides information about the frequency range at which the cross-coherence is maximal at a phase lag. We can examine the activities for known physiological ranges up to 30 Hz, and/or beyond those ranges up to 64 Hz in this case, focusing on different frequency bands to examine different patterns of entrainment for different dyads, etc. (see other examples [24,37].) In this paper, we focus on the activities across all bands. Figure 2D shows the adjacency matrix that we use to represent the connected network, while focusing on the positive phase leads and zeroing out the entries corresponding to negative phase. The OutDegree vector for a construction task time window is shown in Figure 2E and used to represent network connectivity in the third row. There the color bar represents the maximal coherence values in a color gradient used to color the arrows linking the nodes, with direction according to the positive phase value captured by the arrow thickness. The size of the node is the OutDegree value capturing all the links that the node connects to all other nodes. Two states of the network are represented as an example of transitions, whereby the child leads on average in one and the clinician leads on average in the other. Figure 2F reveals other states where different network patterns evolve as the dyad performs the construction task.

*Dyad Coherence Metric*: The total dyad coherence is obtained by summing over the entries of the dyadic portions of the matrix (circled in Figure 2D). The lower left quadrant circled entries represent coherence from clinician-to-child (COH-1) and the upper right quadrant represents coherence from child-to-clinician (COH-2). COH-1 + COH-2 is tracked for all the tasks performed during the session in Figure 2G. Note here that the maximum cross coherence metric is symmetric, but the matrix derived using the phase-lags with positive values to build the adjacency matrix is not symmetric.

*Dyad Strength Index*: An index measuring the strength of the dyad (Equation (3)) combines the value from the Dyad Coherence metric in the numerator and the value from the Dyad Variability metric in the denominator. Higher value of the ratio indicates lower variability and higher cross-coherence during the interaction. This quantity was normalized between (0,1) using the maximal value across all tasks.
(3)$$DyadStrength=\frac{DyadCoherence}{DyadVariability}$$

*Delay in Cohesiveness*: For maximal coherence values, we sum the lag values over the dyadic entries of the phase matrix to determine, under our (*i*, *j*) order convention which person lags behind the other person when the shared dyadic body parts are maximally coherent.

*Social Readiness Metric*: Using the combination of macro-level data (ADOS scores) and micro-level Dyad Strength quantity, we build a parameter space where we localize all controls and autistic participants to learn where in that parameter space the controls lie. The assumption here is that the control with the highest value of Dyad Strength and the lowest value of ADOS score has high social readiness potential. Then we build a relative metric of Social Readiness by taking the difference between this representative participant and every other participant in the cohort. This is a positive quantity spanning from 0 to the maximal value across the cohort. We normalize both the ADOS scores and the Social Readiness quantity using the corresponding maximal values, and we can then profile the full cohort simultaneously at the macro- and micro-level of inquiry. This can inform us about the potential of each child in the cohort for appropriate social interactions that the clinical test expects and that the body performs. 

*Leading Profile:* To assess who leads, the child and clinician, during the dyadic interaction for each task, we use the OutDegree metric (Figure 2E) obtained from the weighted directed graph that the cross-coherence adjacency matrix produces for each task (e.g., in Figure 2D). Given two nodes, the node with the higher value of OutDegree is said to lead the interaction. There are nine possible cases of inter-body connections (3 (child) × 3 (clinician)) in the dyadic network shared by the child and clinician. As the interaction unfolds, these states change (e.g., Figure 2F). At each time unit where we update the state of the network, the maximal OutDegree represents the leading person. Since the total time for every task is different, we calculate the lead-lag dynamics across all the time units in the manner explained above (with 12s windows taken under IID assumption) and sum up the total for each child and clinician. The person leading a greater number of time units for each given task is said to lead the task. Figure 2H shows the lead-lag for every task performed by EP01 during the session. 

As an example of lead-lag calculation for EP01, he spent 17 time units in the Construction Task (CT) with a 12 s time unit (128 Hz × 12 s gives 1536 frames and this is the minimum time length of a task across the entire cohort.) This number of frames provided enough information to empirically estimate statistical parameters with 95% confidence intervals. At time *t* = 1, the OutDegree patterns could be (3 (LW), 3 (RW), 5 (Torso))_Child_ and (1 (LW), 0 (RW), 3 (Torso))_Clinician_. 

Each of the entries in the Child vector are compared to the entries in the Clinician vector, giving a total of nine possible dyadic interactions. For example, the first interaction, being LW_Child_ (OutDegree = 3), is compared to LW_Clinician_ (OutDegree = 1). The Child leads this interaction as it has a higher value of OutDegree. Similarly, the Child leads seven to nine interactions, none are led by the Clinician and two are balanced (with equal OutDegree value). Consequently, at time *t* = 1, the Child leads. 

The lead-lag patterns can be visualized as in Figure 2F, where the color of the lines depicts the coherence value, the line thickness represents the phase lag, and the size of the node is the OutDegree. Here the direction of the arrows represents who leads the interaction at time *t*. We can see that at *t* = 1, the child leads as explained above. The calculation for the other time units follows and a profile across tasks is shown in Figure 2F. Similar to the dyadic interaction portion of the matrix, we can also quantify the self-interaction for each child and clinician separately, to see which sensor (Left Wrist, Right Wrist or Torso) leads during each of the tasks. 

## 3. Results

The combination of macro- and micro-level of inquiries can be appreciated in Figure 3A where we plot the ADOS sub-scores as a function of the Dyad Strength for each child in the cohort.

### 3.1. Potential for Social Readiness Revealed by Micro-Level Socio-Motor Biometrics Is Missed by Macro-Level Criteria

Figure 3A shows that neurotypical children have low ADOS scores (total, social affect, and repetitive restricted behaviors) and higher Dyad Strength than the autistic participants, according to linear acceleration data (angular speed data in Appendix A is congruent.) However, several autistic subjects with high ADOS scores also have strong Dyad Strength, in the range of neurotypicals. Their bodily biorhythms synchronize with those of the rater as they socially interact. This shared level of cohesiveness suggests a potential for social readiness captured by the digital data but missed by the naked eye. To further characterize that potential, we compute the difference between the neurotypical participant with the highest Dyad Strength value and that of every other participant in the cohort. We then normalize this quantity by dividing it by the maximal value of that difference across the group. The corresponding normalized ADOS score is plotted for each participant along with this relative measure of Social Readiness. The closer to 0 this quantity is, the higher the Social Readiness potential. This can be appreciated in Figure 3B inclusive of neurotypical controls and autistic subjects.

Notice that several of the autistic participants have social readiness potential in the range of controls, despite high ADOS scores.

### 3.2. Mirroring of Social Actions Inevitably Biases the Rater 

Another metric that we can derive from the network connectivity analysis is the extent to which each node leads other nodes in the grid of six sensors, conceptualized as an interconnected network. To that end, we use the OutDegree metric (e.g., Figure 2E) which signals the number of links with high cross-coherence from each node to each other node (visualized in Figure 2F.) When examining self-interaction nodes, we focus on the person’s nodes and obtain the OutDegree metric of each node, signaling the phase lead (in degrees per unit time) for each of the maximal cross-coherence values obtained pairwise. We do this for each child in the neurotypical and autistic subgroups and for the rater clinician interacting with these children. Figure 4A shows this profile as % of the neurotypical children’s cohort leading with the highest positive phase at maximal cross-coherence (left panel in 4A). This quantity is plotted for each of the tasks that the children performed for a given module (along the x-axis) across all three sensors. The right-hand panel in 4A depicts the information for the clinician. The neurotypical children and the rater clinician have comparable ranges of leading index values. In contrast, the same clinician shows a very different range of leading index values when the children are autistic. This mirroring effect of Figure 4A,B strongly suggest a socio-motor bias that inevitably occurs largely beneath awareness, but that nonetheless skews the performance of the task by the rater. 

When the interlocutor is neurotypical, the rater’s patterns mirror the range of neurotypical values, but when the interlocutor is autistic, this range shrinks. Since these are self-interactions (inter-node cross-coherence phase for each agent) and the dyad also shares the intra-node cross-coherence, we then obtained graphs in Figure 4C exploring similar a question for each visit. Controls only visit once, whereas autistics come back three more times. In each case, we now examine the OutDegree leading patterns of the dyadic nodes of the matrix. The angular speed data in Appendix A shows consistent patterns with the linear acceleration data of Figure 4.

### 3.3. Systematic Leading Bias of Rater Remains Across ADOS Modules and Familiarity with the Child

We see that, as with the self-interaction nodes, the dyadic interaction nodes reveal a mirroring bias whereby, for the same clinician, same child and same module, the range of values in the leading parameter is on average much higher when the children are neurotypical (Figure 4C, left-panel) but shrinks when the children are autistic (Figure 4C, right-panel). Then, in visit two, when the module changes to an easier one, the range of values broadens across the autistic children. This effect on the person leading the interaction shows the mirroring effects of shared socio-motor parameters that raters inevitably miss when relying on observation alone. 

The same cohort of children returns to visits three and four, but we change the rater. As in visit one, during visit three, the rater administers the most appropriate module for each child. In visit four (as in visit two), the module is the less language-appropriate one. In visit three, the children perform the appropriate module, as in visit one. Despite the familiarity with that module, the patterns of leadership ranges shrink with the new, unfamiliar rater. Here we see that the new rater’s range becomes much broader when the children are familiar with her in visit four, and the module is already familiar to the child, from visit one. For both raters, there is an implicit mirroring effect with the autistic children that is inevitably bound to bias their scoring. The raters are interacting in a closed loop with the child and influencing the interaction that they evoke, respond to and rate. The cross-coherence network connectivity metrics capture the inherent biases in the leading patterns across the ADOS tasks. In all cases the rater dominates the interaction albeit with marked differences between controls and autistics in the full range of values across the cohort, susceptibility to familiarity with the rater and easiness of module.

### 3.4. The ADOS Test Probing Appropriateness in Social Interactions Is Administered as a Monologue

The network connectivity analyses invariably reveal for controls and autistics that the rater leads the interaction most of the time. This can be appreciated in Figure 5A, while Figure 5B shows the summary across all four visits for the autistic cohort and for visit one for the controls. Although natural social interactions of a dyad are supposed to unfold like a dialogue between two agents, the test administration switches the interaction to a monologue style. The rater is unaware of these socio-motor parameters, as they evolve implicitly, from the synchronous cross-cohesive patterns that self-emerge from the two bodies in micro-motion. At the macro-level of the social dyad, these aspects of the dynamically unfolding behavior occur much too quickly to be picked up by the rater, who is busy trying to evoke, react to and rate the child’s social overtures. Appendix A shows consistent results for angular speed data as Figure 5 does for linear acceleration data.

### 3.5. Micro-Level Analyses Capture Socio-Motor Changes Missed by Macro-Level Scores

Figure 6 shows data from a representative autistic experiment participant who returned to the lab for a second visit and was rated by the same clinician during performance of the same ADOS-appropriate module in both visits. We note that the child did not perform all tasks in each visit but performed a subset of them in both visits. While his ADOS scores remained unchanged, his socio-motor patterns markedly changed from visit to visit. Figure 6A shows the patterns of leadership for both visits, while Figure 6B shows the patterns of Dyad Variability. Figure 6C,D show the patterns of Dyad Coherence between child and clinician in the clinician→child and child→clinician directions, respectively. While neurodevelopment continued its complex micro-dynamic course in this autistic participant over two visits separated months apart, at the macro-level the ADOS scores remained deterministically static over time. Appendix A shows consistent results for angular speed data as Figure 6 does for linear acceleration data.

### 3.6. High Sensitivity to Change in Clinician Across All Children of the Cohort

Two raters participated in the study, one in visits one and two, the other in visits three and four. As mentioned, the appropriate ADOS module was administered in visits one and three. Less appropriate modules were administered in visits two and four. This experimental assay allowed us to probe the responses of the dyad to the switch in raters (same ADOS module, different raters.) For the autistic children, the leading profile derived from the self-interaction entries of the matrix changes with the change in clinician, despite maintaining the same ADOS module, whether appropriate or not (Figure 6E,F). This suggests that the child’s nervous system is sensitive to changes in the social environment, a desirable feature that could be further used to enhance social interactions across different situations.

### 3.7. Stratification of Sub-Types in Autism Is Not Possible Using Macro-Level Criteria Alone

Among the experimental participants, several had similar ADOS scores. This is not surprising, given the narrow numerical range of discrete values of these scores and the large heterogeneous population sample. Upon taking a random draw of participants from the population, it is not possible to cover all different subtypes using these discrete scores’ range from macro-level criteria. To instantiate this point, we present the socio-motor data derived from the biosensors for two experiment participants with identical ADOS scores but fundamentally different socio-motor phenotype. We examined their ADOS activities using the same module and rater. Figure 7A shows the leadership profile across tasks for these two representative autistic participants with very different socio-motor activity. At the macro-level of ADOS description they are similar. At the micro-level of socio-motor behavior they differ. In an interventional therapy following this ADOS detection, these children would likely require different approaches that such discrete coarse clinical criteria could not differentiate. Panel 7B further shows the distinct profiles of Dyad Variability for these participants, while panels 7C and 7D show the profiles of Dyad Coherence across tasks, taken from the clinician→child and child→clinician dyadic entries of the cross-coherence matrix representing the interconnected network of sensors. Appendix A shows consistent results for the angular speed data as Figure 7 does for linear acceleration data.

### 3.8. Two Different Clinicians Perceive the Same Child’s Socio-Motor Patterns Highly Differently 

Likewise, following up the sensitivity of the children’s nervous system to changes in clinician, we here asked if the same autistic child interacting with the two clinicians in different visits would be differentially perceived by the clinicians. We tested this question using the micro-level socio-motor patterns from the clinicians and comparing them across the tasks that were common to the child. Figure 7E shows the differences in leadership patterns of the clinicians for the same child while Figure 7F shows the profile of Dyad Variability. Figure 7G,H show the pattern of Dyadic Coherence for the clinician→child and child→clinician directions, respectively.

### 3.9. Task Ranking and Target Treatment

Given the sensitivity of micro-level socio-motor patterns to changes in clinician, modules and the passage of time from visit to visit, we then ascertain the extent to which we could use the digital data derived using the ADOS test as an outcome measure and possibly turn this test into an intervention tool, beyond the static notion of a macro-level detection instrument. Adding the socio-motor micro-dynamics is amenable to further investigating this question, task by task. To that end, we probe the delay in attaining cohesiveness as a metric of natural social interactions. Delay in Cohesiveness (see methods) is obtained by summing over phase-lag values of the entries of the phase matrix for which maximal cross-coherence was attained between the child’s and the clinician’s biorhythmic activities. This is the socio-motor space cohesively shared by the two agents of the dyad when their bodily biorhythms synchronize with different phase lags. At 0 lag they are perfectly in sync, but that state is often transient. Lag patterns add up to out-of-phase synchronous patterns across child and clinician. We measure that Delay in Cohesiveness and build a parameter space to learn about this as a function of Dyadic Strength (previously defined.) We find that controls span the full range of Delay in Cohesiveness while most lie on the higher values of Dyadic Strength (Figure 8A). In contrast, autistics are mostly on the upper left quadrant of the graph, with high values of Delay in Cohesiveness and, on average, lower values in Dyadic Strength than controls.

This result motivated us to examine the tasks individually and propose a task training regime whereby, in a personalized manner, one could rank which tasks have highest Dyad strength, with variable delay in cohesiveness that converges toward the right lower quadrant of the graph. Then we can use those tasks to start gradually training the person while incorporating other tasks, as social learning improves in the precise sense that Dyad Strength increases, Delay in Coherence is variable and, at the macro-level, the rater begins to unambiguously perceive the positive changes. Figure 8B provides an example of the continuous digital scale derived from the tasks, ranging from lowest to highest Dyad Strength value. These scales encompass both macro- and micro-level of information but is dynamic, as it depends on the evolving socio-motor skills of the person interacting in a dyad. Appendix A shows the tasks that are significantly different at the alpha of 0.05 when comparing controls and autistics. Appendix A shows consistent results for angular speed data as Figure 8 does for linear acceleration data.

### 3.10. The Digitazed ADOS Separates Males and Females Using Network Connectivity Metrics and Noise to Signal Ratio (NSR)

The network connectivity analyses revealed fundamental differences between males (16) and females (10) in the cohort. Figure 9 shows the distributions and scatters of several parameters used to assess dyadic socio-motor behavior. For each parameter we used the same number of points, 1730, which was the minimum set in the females derived from the 27 tasks. We cycled across all participants of the cohort and sampled multiple sets to systematically compare males vs. females. The sets represent a random draw of the population, inclusive of autistics and controls, whereby females revealed a very different connectivity and NSR profile than males. All parameters were statistically different with significance at the 0.01 level, *p* << 0.01 according to the non-parametric rank-sum test.

The network connectivity parameters revealed better connectivity patterns in females than males. Since these patterns are derived from the dynamic dyad, they imply better social rapport with the clinician in females than males. The characteristic pathlength was significantly shorter in females than males, suggesting faster information transmission throughout the network, a feature that was accompanied by significantly lower NSR in females, lower clustering coefficient values denoting more distributed network in females, with more efficient (sparser) dyadic node information accompanied by higher betweenness centrality in the dyad nodes. These patterns turned out to be highly different when comparing males’ and females’ micro-movement data, as generated by the dyadic network from the digital ADOS. However, better connectivity patterns in females may make it harder for the clinician to visually recognize social difficulties in the female phenotype. All comparisons of these parameters yielded highly significant differences for the time series data from the accelerometer (Figure 9A,B) and from the gyroscope (Figure 9C,D).

## 4. Discussion

This study leveraged the digital data output by wearable sensors during the performance of a standard clinical test to significantly augment the information that we could gain during the assessment of dyadic social interactions. Turning the standard tasks of the research ADOS test into experimental assays and manipulating several parameters of the experiment, we were able to improve the characterization of socio-motor behaviors across the population, inclusive of neurotypical controls and autistic participants.

We digitized this test while neurotypical children interacted with the rater and uncovered new aspects of the test that helped reveal the potential for social readiness in autistic children. We also learned that the rater dominates the interaction, leading most of the time in both neurotypicals and autistics, with a mirroring effect that inevitably biases the shared cohesiveness of the interaction, channeling a broader range of variability in leadership patterns for neurotypicals and a much narrower range in autistics. The mirroring effects occur at the personal level of activity in what we coin self-interactions of the nodes (conceptualized as kinematic synergies across the bodily biorhythmic fluctuations of each interlocutor in the dyad.) It also occurs at the shared spatio-temporal patterns of cross-coherence between several body parts of the child and the rater. This shared cohesiveness that self-emerges and evolves across the tasks in a session contributes to the strength in dyadic interactions present in autistic children and is therefore revealing of their potential for social readiness. We would not have discovered this socio-motor parameter had we not included neurotypicals in our research-ADOS assessment. As social interactions depend on developmental socio-motor milestones, and these in turn depend on the maturation of the nervous system, our results underscore the need to include normative data from neurotypical participants in any study or assessment of autistic social behaviors. Furthermore, objectively quantifying the level of bias from the rater during the interactions enables us to curtail it or enhance it as needed, thus providing new means to causally induce a child’s reaction in a much more controlled manner than has been traditionally done.

The use of the research ADOS standardized modules in our study allowed us to test the same child with the same rater under different social demands. When the module was appropriate, we saw a consistent reduction in the range of variability in the leading patterns across raters, in relation to their patterns for a less appropriate module under a context of more familiarity with the rater. This is important because, at the micro-level of socio-motor behaviors, the same child may demonstrate more social competence, given that s/he is more familiar with the person and/or the tasks. This is not something apparent to the naked eye of an observer and much less so when the observer may be also distracted when trying to rate the interaction. There is a bottleneck in information processing that inevitably prevents the rater from seeing this aspect of the interaction. Adding the micro-level of analysis thus presents us with the opportunity to harness this potential for social readiness and offer the child (autistic or otherwise) better social support. We can then utilize this standardized test as an avenue to train the child to be social without having to instruct how to do so, but rather by leveraging this micro-layer of information that we can extract from the shared spatio-temporal cohesiveness of the dyad under more relaxed conditions (e.g., more familiarity with the rater and the task.) Socio-motor phenomena are adaptable. We can use these biometrics to track the progression of adaptation at the micro-level of inquiry, underscoring once again how much more we gain by integrating them into the current behavioral assessments tools at the research and at the clinical levels.

Dyadic social interactions are typically a dialogue with balanced contributions from both parties involved. They succeed at communicating when there is appropriate turn taking, when joint attention is not sustained for too long or when it is not too brief, and when the entrainment between the bodily biorhythms lead to implicit agreement supporting non-verbal communication. There is a delicate balance at the micro-level activity that scaffolds social rapport of the interacting dyad at the macro-level. The results from our analyses reveal a prevalence of the rater leading the interaction with both neurotypicals and autistics. This rater dominance turned the test into a quasi-monologue style of interaction, perhaps an unintended consequence of having the same person evoke the social overture of the child, while responding to it, and rating the outcome. Inevitably, the leadership role of the rater skews the interaction. Although this is not discussed in the social behavioral sciences, here we reveal it using micro-level analyses of the shared spatio-temporal and frequency parameters, characterizing the fluctuations of the biorhythmic activity generated by the two bodies in motion during the dyadic exchange.

This strong skewing influence of the rater on the dyadic social behaviors changes the outcomes of the child’s socio-motor patterns for the exact same task and module when different clinicians administer the research ADOS to that child. Consequently, not only at the micro-level of social responses, but also at the macro-level of observable responses, a given child could be perceived very differently by different clinicians. There are both positive and negative consequences here. The positive side is that we can precisely characterize the child–rater’s reactions to each other and, in the context of therapy, use this digitized instrument to determine which therapist has by default more rapport with the child. This would be especially relevant in the early stages of the therapy. Likewise, we could use this test and biometrics in the context of a teacher-child relationship, to select the teacher that will most likely (by default) have the best rapport with the child. This is important for emotional social learning in general.

The negative aspect is that, for a test that aims at detecting something with high precision, there is no chance that it will do so. The detection problem becomes ill-posed: given a score, what is the most likely socio-motor phenotype? The map from score to socio-motor phenotype is many-to-one. This result bears implications for our (in)ability to define the autistic phenotype(s) and reliably uncover autistic subtypes in a random draw of the population. We could not use this test to stratify the broad spectrum of autism.

Since this test informs and largely steers the science of autism, and since given the ADOS score it is not possible to infer the most likely socio-motor phenotype, for detection purposes it may be beneficial to combine different levels of inquiry and produce a more reliable objective metric impervious to rating styles, personality or cultural nuances. This should be possible by providing a digital characterization of social interactions in general, as we have done here, with a clinical example that uses standard pre-specified tasks amenable to illustrate these various points.

We emphasize that there are many more possible biometrics that we can derive from these tasks and interactions [37,38,39,40]. The examples used here are merely illustrative of the broad variety of socio-motor parameters that digital data can offer. Wearable biosensors today sample at high frequency even when they are off-the-shelf, such as smart phones, smart watches, etc. [41] affording science and clinical areas a new level of granularity sensitive to the detection of change and its rate during neurodevelopment. This information is now at the tip or our fingers owing to the wearable sensors revolution, offering continuity to behavioral science even in times of physical (social) distancing.

One important aspect of the study was that some of the autistic children returned to the lab for a total of four visits. This passage of time was important to assess the extent to which the socio-motor phenomena assessed at different levels of inquiry would reflect the rate of change in neurodevelopment that the children underwent. We found that while the macro-level of description from observation (using in this case the research ADOS scoring criteria) remained static, the micro-level of description and statistical inference fluctuated over time, describing a stochastic trajectory that could reveal individual trends for each child and for the cohort.

Having the ability to assess developmental trajectories in tandem with changes in somatic-sensory-motor physiology is important to build the notion of a dynamic diagnosis chart that could reflect such changes in maturation and function. Underlying all these socio-motor biorhythmic phenomena is a growing, adapting and developing (coping) nervous system. Thus, relying exclusively on coarse observational metrics that fail to capture change will prevent the ability to provide proper support to nurture the growth of these systems and the social axes that they scaffold. Introducing these digital biometrics of socio-motor behaviors can help us bridge the gap between neurodevelopmental rates of change and coarse clinical behavioral criteria.

Another aspect of the study that is worthwhile underscoring is the fundamental differences in network connectivity patterns that we found between males and females. These differences in connectivity patterns relate to socio-motor parameters derived from the dynamically interacting dyad. Thus, unlike the paper-and-pencil version of the ADOS, which has failed to detect social differences in females (by observation), here the digital ADOS does so across all parameters under consideration. Furthermore, we also examined the Gamma NSR empirically derived using MLE and found fundamental differences between the MMS signatures of males and females across the cohort. This result is congruent with prior work involving person-centered analyses of involuntary head motions [32,42,43], decision making by pointing to targets [44] and gait analyses [45] from our laboratory. The new result separating sex in autism is more relevant than the previous results reported because it is derived from the dyadic social interaction taking place during the very tasks that define the scores to detect the condition. These results mean that, across numerous socio-motor axes, there is a high likelihood of automatically (and blindly) detecting such sex differences in a random draw of the population performing the ADOS test. This randomly chosen cohort of 26 children opens a new line of inquiry with the potential to finally address the disparate 4-5:1 male to female ratio in autism [46,47]. Most important yet, we will be able to do so using the test that researchers and clinicians use to detect autism, if we digitize it. Because of the wearable sensors revolution, we can easily digitize the ADOS using off-the-shelf technology such as smart watches and web-cameras, an important advantage also in remote testing and telemedicine approaches due to the pandemic. Under these new conditions, clinicians have limited visibility from the point of view of the child-caregiver dyad at the other end of the screen. The connectivity metrics and the empirically derived Gamma-NSR are hidden to the naked eye, but visible to the digital data and to the analytics that we offer here. It will be important to scale up this result and validate it across thousands of dyads performing the ADOS test while capturing the dynamic dyad digitally. We here note that our methods extend to more than two people, so remote model of telemedicine involving the clinician at one end and the parent-child dyad at the other end is also feasible.

Adopting standardized digital technology and integrating it with these gold-standard psychological tests may help us develop new outcome measures of treatments and provide parameter spaces where we can track the trajectories of interventions. A testable hypothesis is that, as participants repeat the ADOS performance, it may be possible to ascertain which tasks give higher dyad strength. If we were to focus on those tasks first to improve social interaction, then it would be possible to shift to the other tasks, thus aiming at broadening the repertoire of tasks that lead to appropriate social exchange, tailoring this to the child’s inherent strengths in an individualized manner. Indeed, digitizing social behaviors could provide a guide to assess the changes in socio-motor parameters introduced here such as Dyad Strength and Delay in Cohesiveness, computed as relative metrics, obtained first in neurotypicals, and as such revealing of the expected course of socio-motor development, constructed along the path of least resistance. Our work offers more than one way to do this while leveraging the wearable sensors revolution and preserving the clinical gains already attained with observational means.

### Caveats and Limitations of the Proposed Hybrid Methods

Although the digital characterization of the ADOS test may hold the promise of revealing new aspects of the ADOS interactions that may otherwise remain hidden to the naked eye, there are several caveats regarding the adoption and use of the present methods. The major limitation is the current lack of mechanisms that can scale up the results from this reduced cohort in the laboratory to thousands of participants in clinical settings. Future use of the analytics presented here can be extended to biorhythms registered with off-the-shelf wearables such as smart watches and web cameras outputting shareable digital data, even across remote settings. To that end, our laboratory has developed a new concept “The Lab in a Box”. This remote kit for behavioral data acquisition and analyses offers continuity to the administration of behavioral tests such as the ADOS in times of social (physical) distancing. However, deploying such methods on a large scale would require a concerted effort across multiple disciplines, business models and entrepreneurship beyond academic settings. This exceeds the scope of this paper, but it is perhaps something to keep in mind in order to translate the present results to clinical and home settings, offering the possibility to scale up the use of this technology to tens of thousands of users.

Another limitation of the present analysis is that other pipelines could have been used to analyze the data. This begs the question of whether the general results would have dramatically changed in that case. In earlier preliminary analyses of these data sets using different analytical pipelines, larger number of sensors in the interconnected grid, different bodily locations for the additional sensors, different kinematic parameters, and different children, we observed similar trends [24]. As such, we are confident that the trends would remain invariant to such nuances. However, we do acknowledge that the interpretation and inference are to be discussed and validated across the community, to provide a consensus on the clinical interpretability of digital data in general. In other words, the characterization of the digital ADOS data that we provide here needs to be further validated across thousands of participants and clinicians. This is the case for any new scientific approach. In this sense, we invite the reader to reproduce the results using ours and other data sets provided in GitHub by our laboratory. Our analytics do not make theoretical assumptions about the likelihoods and probabilities of representing data features. Our new methods, rather, empirically estimate these from the micro-fluctuations inherently present in the signals that we register from natural behaviors. In this sense, we remain confident that cross-sectional and longitudinal data captured with various means will find trends similar to those found in this random draw of the population of autistics and neurotypical controls. This work extends an invitation to the clinical community, as well as to the scientific community, to collaborate and take steps toward combining different levels of inquiry in the study of general social behaviors beyond autism and other neurodevelopmental conditions.

## 5. Conclusions

We have shown that by integrating macro-level rubrics used to assess appropriateness of social exchange with micro-level biorhythmic activities co-registered synchronously in both agents of the social dyad, we can uncover much more than we would when using one level of inquiry alone. Our work underscores the need to combine pencil and paper methods with objective instruments that add a finer layer of granularity to the study of the complex dynamics of social phenomena. Without a doubt, this is a case where the overall sum amounts to much more than its individual parts.

## 6. Patents

EBT holds the US Patent “Methods and Systems for the Diagnoses and Treatments of Nervous Systems Disorders” combined in the paper as micro-movement spikes, MMS data type and Gamma process.

## Figures and Tables

**Figure 1 jpm-10-00159-f001:**
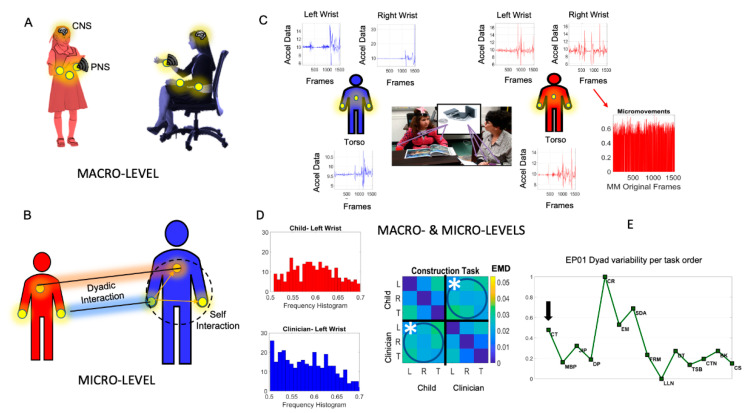
Leveraging the wearable sensors revolution to significantly augment traditional pencil and paper methods and advance the behavioral sciences. (**A**) The macro-level of behavioral description inevitably misses fast and subtle information in dyadic social interactions, as information flows between the Central and the Peripheral Nervous Systems (the CNS and the PNS). (**B**) Micro-level coherence information, automatic social mirroring, lead-lag patterns, and micro-gestures of the face and body, among other socio-motor behaviors, can only be captured with high grade instrumentation and proper analytics. (**C**) Macro- and micro-levels of inquiry can be integrated to advance social behavioral sciences, e.g., the Autism Diagnostic Observation Schedule (ADOS) is used as a backdrop experimental assay to probe dyadic social interactions, combined with wearables. to digitize standard clinical criteria. Light wearables embedded in the clothing unobtrusively and continuously co-register child and clinician (certified as rater) during the administration of the ADOS test to detect autism. Micro-fluctuations in the timeseries data from these wearables (e.g., acceleration waveforms) can be converted to standardized micro-movement spikes (MMS) to derive various socio-motor biometrics. (**D**) The Earth Mover’s Distance (EMD) metric is used to ascertain the pairwise difference between frequency histograms of MMS and build a matrix with entries capturing the dyadic interactions (circled entries) and the self-interactions of body nodes. (**E**) The averaged sum across all dyadic interaction entries per unit time, obtained task by task, in the order in which they were administered, gives us a session profile of the dyadic variability, as one example of several metrics used here to objectively quantify socio-motor behavior simultaneously at the macro- and micro-levels of inquiry.

**Figure 2 jpm-10-00159-f002:**
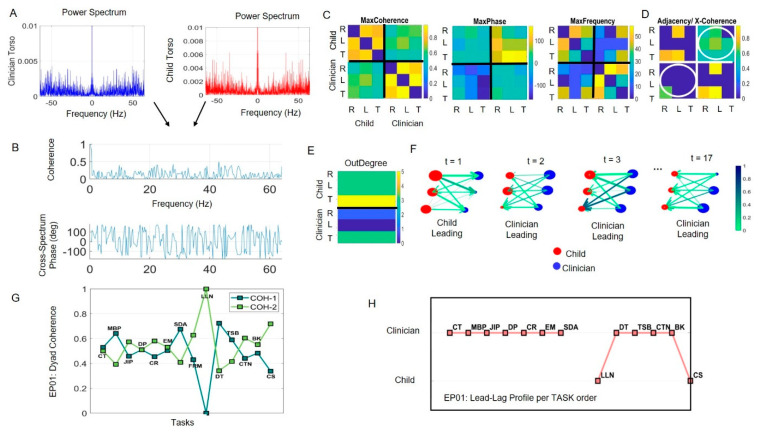
Analytical pipeline and visualization tools. (**A**) Frequency domain power spectrum analysis of pairwise sensor data provides input to obtain cross coherence spectrum phase across frequencies. (**B**,**C**) A parameterization of the data into three corresponding matrices: MaxCoherence, MaxPhase, MaxFrequency. Each entry contains the maximal cross coherence value obtained pairwise across all sensors. For example, row 1 contains the values for the child’s right wrist (R) sensor in relation to all other sensors (child left wrist L, child torso T, clinician right wrist R, clinician left wrist L, and clinician torso T. Dyad entries of the matrix are at the right upper quadrant (child→clinician) and left lower quadrant (clinician→child). Self-interaction entries are at the left upper quadrant (child) and the right lower quadrant (clinician). Corresponding entries in the MaxPhase matrix reflect the phase value at which the pairwise coherence is maximal while the same entry in the MaxFrequency reflect the corresponding frequency value. (**D**) Adjacency matrix used to represent a dynamically changing weighted connected graph is obtained by retaining the entries at positive phase under convention i→j for the (i,j) entry. Dyadic interaction entries are circled. (**E**) Outdegree for each node denotes the number of links from that node to other nodes. (**F**–**H**) Visual tools and sample metrics to characterize socio-motor behaviors in social dyads. (**F**) Visualizing the network to track its states as they dynamically change in the Construction Task for a child-clinician dyad (EP01, with 17-unit times of 12 s time length for each sample under Independent Identically Distributed (IID) assumption, explained in Methods). The size of the node corresponds to the OutDegree value, the arrow’s color is the maximum cross-coherence value, the thickness is the phase value and the direction comes from the computation involving Outdegree (see methods.) (**G**) Sample profile over tasks of Experimental Participant 01, EP01 showing the COH1 and COH2 unfolding within the session. These terms are used to obtain a socio-motor metric (see Methods.) (**H**) Lead profile for the dyad involving EP01 and revealing the clinician prevalent leadership for most tasks, but Loneliness (LLN) and Creating a Story (CS), where the child leads (please see Appendix A for task description and acronyms).

**Figure 3 jpm-10-00159-f003:**
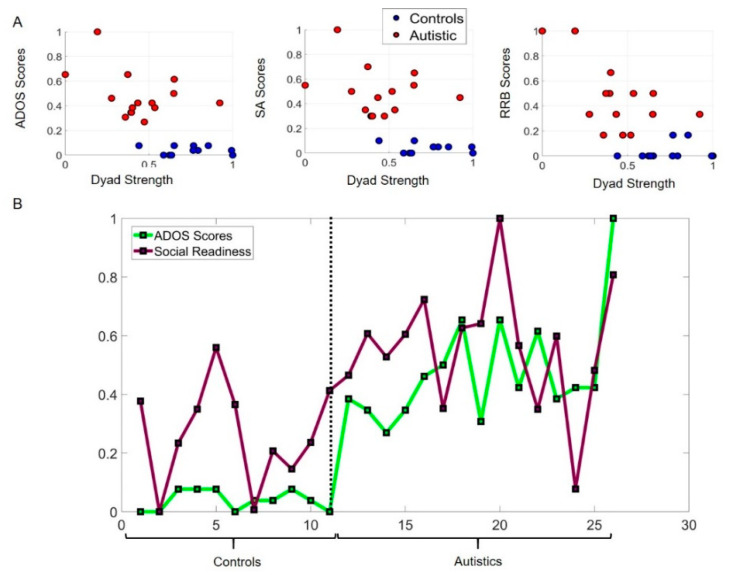
A measure of social readiness potential: Dyadic strength expressed in relation to ADOS sub-scores (Social Affect, (SA) and the so-called Repetitive Ritualistic Behaviors, (RRB)). (**A**) Dyad strength (see methods) tends to be higher in neurotypical controls, with a tendency towards lower (normalized) ADOS scores and some higher variations in the more ambiguous RRB sub-score. The trend in the autistic dyadic strength scores is towards lower values and higher ADOS score, yet 5/15, (33.3%) fall within the neurotypical lower range despite higher ADOS scores, thus signaling a hidden capacity for social dyadic exchange. (**B**) This information can be further unfolded for each participant, whereby a score of social readiness potential is obtained as a relative quantity measuring the absolute difference between each participant and the neurotypical with the highest dyad strength. Many of the autistic participants do have socio-motor strength in the dyadic interaction, despite high ADOS scores.

**Figure 4 jpm-10-00159-f004:**
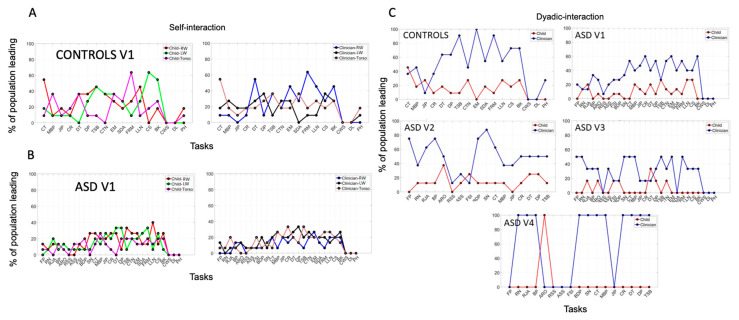
Mirroring Bias Effects. (**A**,**B**) Biased rating in the ADOS test quantified through mirroring metric of lead-lag social patterns of self-interactions (individual kinematic synergies) as percentage in leading patterns across the cohort. Neurotypical Controls and Clinician show broader range of values than Autistics the visit one, under the most appropriate module and same rater. (**C**) Mirroring effect and raters’ leading bias persist in the shared dyadic cohesiveness as the module and rater change. Visit one and three (appropriate module) under two different raters show a reduction in parameter range for autistics relative to controls. Visit two and four (less appropriate module and by then familiar rater) show an increase in parameter range for autistics relative to visits one and three.

**Figure 5 jpm-10-00159-f005:**
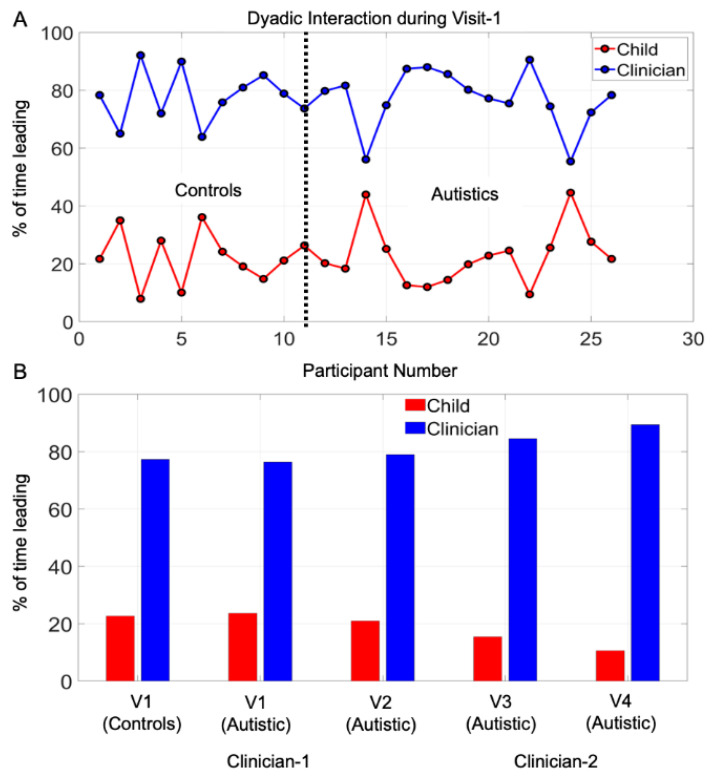
Monologue style of ADOS test. Rater leads social interaction a large percentage of the time for each child. (**A**) For all 26 children divided into neurotypical controls and autistics on the x-axis and on the y-axis, the % of time (taken across all ADOS tasks) that the rater or the child leads the interaction. Clinician rater leads on average for each child, across all tasks. (**B**) Group data per visit showing the summary of the % time that the person leads the social interaction.

**Figure 6 jpm-10-00159-f006:**
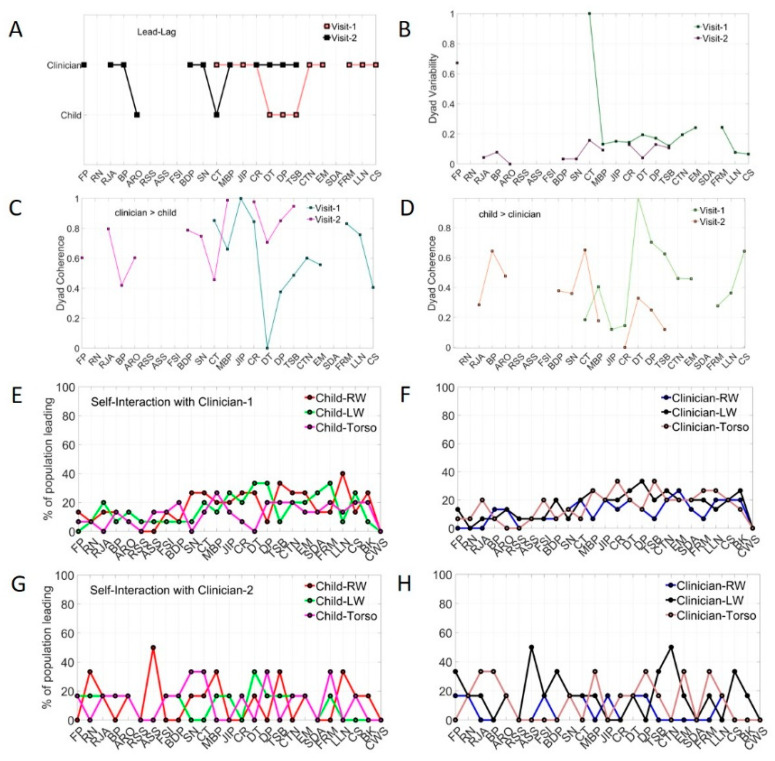
High sensitivity of micro-level metrics captures socio-motor changes over time and serves to measure rater’s reliability-style. (**A**–**D**) ADOS scoring system does not capture change in developmental socio-motor physiology. Across visits, the macro-level scores remain static, but the micro-level socio-motor parameters change. These include leading profile, Dyadic Variability and Dyadic Coherence. (**E**,**F**) High sensitivity to changing clinician rater prevails across all children and tasks. The self-interaction parameters (individual kinematic synergies) of autistic participants are sensitive (at the micro-level) to changing the rater. (**G**,**H**) Leading profiles in the same autistic cohort shift as the raters differ.

**Figure 7 jpm-10-00159-f007:**
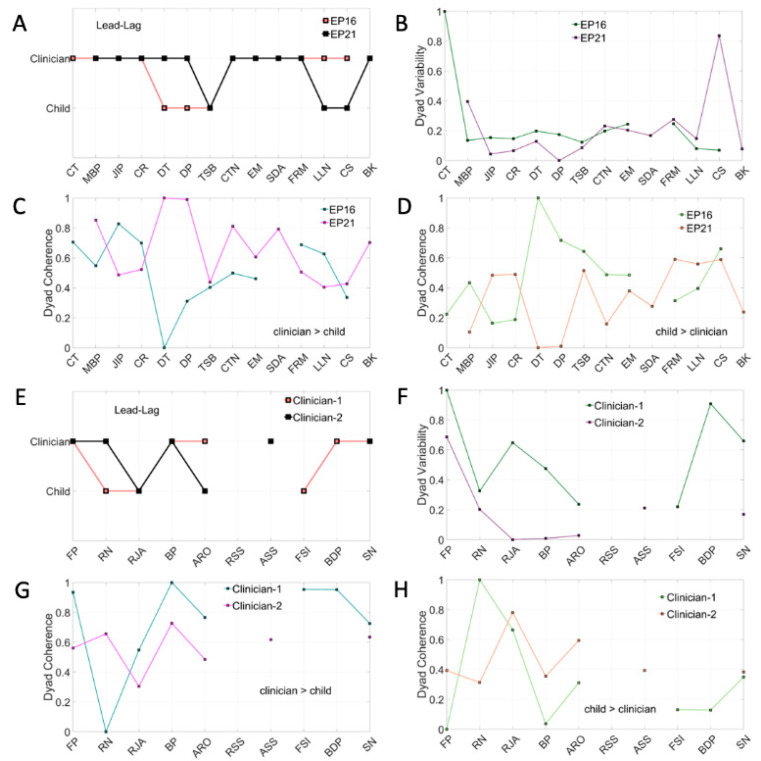
Vignettes samples show bottlenecks of pencil and paper observation methods. (**A**–**D**) Macro-level scoring system does not capture change in developmental socio-motor physiology. Experimental participants EP16 and EP21 are perceived very different by the same clinician in relation to leading patterns (**A**); Dyad Variability (**B**) and Dyad Coherence (**C**,**D**) in each of the clinician→child and child→clinician directions. (**E**–**H**) Ill-posed autism detection problem: Given the ADOS score, what is the most likely socio-motor phenotype? Each clinician perceives the same child differently across all micro-level socio-motor indexes.

**Figure 8 jpm-10-00159-f008:**
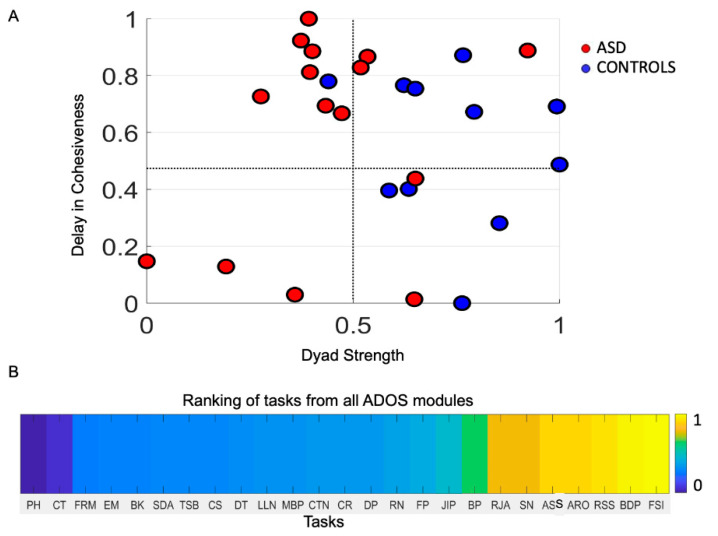
Digital biomarker for personalized design of adaptive targeted therapy. (**A**) Parameter space spanned by dyad strength on x-axis and delay in cohesiveness (see text for details) on the y-axis. Ideally within a social interaction one would desire high dyad strength and a broad range of response cohesiveness, spanning from fast to slow. Note that most controls have high dyad strength and their responses vary broadly from lower to higher cohesiveness delay, whereas autistics are primarily in the region of low dyad strength and delayed cohesiveness (i.e., they synchronize their body biorhythms with those of the rater, but there is a larger lag than desirable.) (**B**) Task ranking criteria based on dyad strength calculated for various tasks from participants in visit one. Functional and Symbolic Imitation, FSI is assigned the highest rank (easiest task to perform) as it depicts the best dyad strength while PH having the lowest dyad strength is assigned the lowest rank (most difficult task to perform).

**Figure 9 jpm-10-00159-f009:**
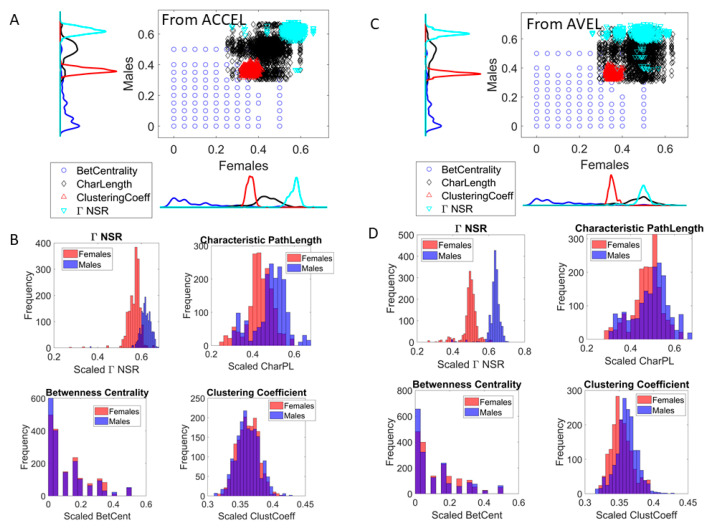
Digital ADOS automatically separates females from males in a random draw of the population. (**A**) Scatters and histograms of parameters derived from interconnected networks representing the dynamic dyad provide evidence for fundamental differences in connectivity across the nodes of the network composed by the child and clinician dyad. (**B**) The empirically estimated Gamma scale parameter (the noise to signal ratio, (NSR)) of the micro-movement spikes (MMS) derived from acceleration separate females from males with lower NSR in females and tighter distribution than males. The characteristic pathlength denoting average shortest distance path from each node to every other node of the network is significantly shorter in females than in males. The betweenness centrality is higher in females than males and the clustering coefficient is also higher in females. All differences are statistically significant with *p* << 0.01. (**C**) Summary graph for angular speed is consistent with the acceleration patterns. (**D**) Females show lower NSR, lower characteristic pathlength and higher betweenness centrality and lower clustering coefficient. All differences are statistically significant according to non-parametric rank-sum test *p* << 0.01.

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
