# Peer review of "Digitized ADOS: Social Interactions beyond the Limits of the Naked Eye"

_jpm, 2020, doi:10.3390/jpm10040159_

Round 1

Reviewer 1 Report

This is a well written paper exploring the possibility of improving the study of social interaction by combining traditional clinical tools with digital data. Furthermore the author proposes that combining these “micro” and “macro” levels of investigation can be usefull to distinguish subtypes of autism which in turn will be usefull when designing treatments.

A strength of the study is the combination of data from “non-autistic” children with autistic children.

The paper is generally well-written with clear explanations.

I do however lack data on the test subjects (both non- autistic as well as autistic children)  such as gender, IQ level). It would be helpful if this could be added. The sample size is small but maybe it would be interesting to see if there are gender differences.

Author Response

Comments and Suggestions for Authors

This is a well written paper exploring the possibility of improving the study of social interaction by combining traditional clinical tools with digital data. Furthermore the author proposes that combining these “micro” and “macro” levels of investigation can be usefull to distinguish subtypes of autism which in turn will be usefull when designing treatments.

A strength of the study is the combination of data from “non-autistic” children with autistic children.

The paper is generally well-written with clear explanations.

Comments

I do however lack data on the test subjects (both non- autistic as well as autistic children)  such as gender, IQ level). It would be helpful if this could be added. The sample size is small but maybe it would be interesting to see if there are gender differences.

We thank the reviewer for the very useful comments and address the comments below

On the Table S3, we have added the sex column to denote the sex of the participant. However, we do not have the IQ scores. Below we corrected our text to be more specific of sex, age and inclusion/exclusion criteria.

We tested a total of 29 participants, 10 females and 19 males (see Table S3). There were 11 control children of school age (mean age 9.9 years old) and 18 autistics (mean age 8.3 years old). In this set 26/29 dyads had complete digitized ADOS with 6 synchronous sensors, 3 on the child and 3 of the clinician co-registering their motions in tandem) and these were included in visit one. The 3/29 dyads not included in these analyses had grids with 12 sensors and/or the child had a diagnosis of Fragile X. They will be the focus of a different publication, so we include only idiopathic autism in the present work. Furthermore, 8 autistics returned in visits one and two, six returned to the lab in visits one through three and one in visits one through four. The details of these participants, along with the module delivered and ADOS-2 score assigned across all the visits are summarized in Tables S2-S3 of the Supplementary Material. Autistic children were referred to our laboratory by local clinics and by the community, with help from the New Jersey Autism Center of Excellence (https://njace.us/), funded by the New Jersey Governor’s Council for the Medical Research and Treatments of Autism.

Sex Differences:

Yes, we examined the females vs. the males. We find statistically significant differences p<<.001 across all parameters (in the accelerometer and gyroscope data) between boys and girls in the cohort across multiple socio-motor parameters of motor variability and network connectivity. These include the noise to signal ratio (the Gamma scale parameter) derived from the micro-movements (fluctuations) of the acceleration/angular speed; the connectivity parameters (modularity, clustering coefficient, betweenness centrality and characteristic pathlength). We now devote a section of the methods to explain these connectivity metrics and aid interpretation/inference.

Among the connectivity metrics that we used, we specifically assessed possible differences between males and females across the cohort. We used the clustering coefficient metric a measure of the degree to which nodes in a graph tend to cluster together. We used the Betweenness Centrality metric, measuring the extent to which a vertex lies on paths between other vertices. Vertices with high betweenness may have considerable influence within a network by virtue of their control over information passing between others. Related to this, we examined the characteristic pathlength, the average shortest distance path (geodesic) denoting he number of edges in the shortest paths between all vertex pairs. In all cases we gathered the data generated by the dyad across the tasks and used the same number of data points in males and in females, to ask if given the scatter of similar size, the values distributed significantly differently. Since we had fewer females (10) than males (16), we relied on the minimum number of values in the females and cycled through all the values of the males (bootstrapping from the data) to ascertain the degree to which they differed when the number of elements were identical for each parameter and for each cohort. To compare the groups, we used the ranksum test and visualized the histograms and the scatters across all connectivity parameters scaled to range between 0 and 1. This normalization allowed us to visually compare males and females across network connectivity metrics and stochastic parameters.

We added the results from these analyses

3.10 The Digitazed ADOS Separates Males and Females Using Network Connectivity Metrics and NSR

The network connectivity analyses revealed fundamental differences between males (16) and females (10) of the cohort. Figure 9 shows the distributions and scatters of several parameters used to assess dyadic socio-motor behavior. For each parameter we used the same number of points, 1,730 which was the minimum set in the females derived from the 27 tasks. We cycled across all participants of the cohort and sample multiple sets to systematically compare males vs. females. The sets represent a random draw of the population, inclusive of autistics and controls whereby females revealed a very different connectivity and NSR profile than males. All parameters were statistically different with significance at the 0.01 level, p<<0.01 according to the non-parametric ranksum test.

The network connectivity parameters revealed better connectivity patterns in females than males. Since these patterns are derived from the dynamic dyad, they imply better social rapport with the clinician in females than males. The characteristic pathlength was significantly shorter in females than males, suggesting faster information transmission throughout the network, a feature that was accompanied by significantly lower NSR in females, lower clustering coefficient values denoting more distributed network in females with more efficient (sparser) dyadic node information accompanied by higher betweenness centrality in the dyad nodes. These patterns turned out to be highly different when comparing males and females’ micro-movement data, as generated by the dyadic network from the digital ADOS. However, better connectivity patterns in the females may make it harder for the clinician to visually recognize social difficulties in the female phenotype. All comparisons of these parameters yielded highly significant differences for the time series data from the accelerometer (Figure 9A-B) and from the gyroscope (Figure 9C-D).

Figure 9.  Digital ADOS automatically separates females from males in a random draw of the population. (A) Scatters and histograms of parameters derived from interconnected networks representing the dynamic dyad provide evidence for fundamental differences in connectivity across the nodes of the network composed by the child and clinician dyad. (B) The empirically estimated Gamma scale parameter (the noise to signal ratio, NSR) of the MMS derived from acceleration separate females from males with lower NSR in females and tighter distribution than males. The characteristic pathlength denoting average shortest distance path from each node to every other node of the network is significantly shorter in females than in males. The betweenness centrality is higher in females than males and the clustering coefficient is also higher in females. All differences are statistically significant with p<<0.01. (C) Summary graph for the angular speed is consistent with the acceleration patterns. (D) Females show lower NSR, lower characteristic pathlength and higher betweenness centrality and lower clustering coefficient. All differences are statistically significant according to non-parametric ranksum test p<<0.01.

We also devote a paragraph in the Discussion to place the results within the context of prior work involving various motor control parameters. We had found already motor parameters indicating such differences in autism from voluntary and involuntary motions but never within the dyadic context. We revisit these results too in the discussion.

Another aspect of the study that is worthwhile underscoring is the fundamental differences in network connectivity patterns that we found between males and females. These differences in connectivity patterns relate to socio-motor parameters derived from the dynamically interacting dyad. Thus, unlike the paper-and-pencil version of the ADOS, which has failed to detect social differences in females (by observation), here the digital ADOS does so across all parameters under consideration. Furthermore, we also examined the Gamma NSR empirically derived using MLE and found fundamental differences between the MMS signatures of males and females across the cohort. This result is congruent with prior work involving person-centered analyses of involuntary head motions [32, 42, 43], decision making by pointing to targets [44] and gait analyses [45] from our lab. The new result separating sex in autism is more relevant than the previous results that we reported, because it is derived from the dyadic social interaction taking place during the very tasks that define the scores to detect the condition. These results mean that across numerous socio-motor axes, there is a high likelihood of automatically (and blindly) detecting such sex differences in a random draw of the population performing the ADOS test. This randomly chosen cohort of 26 children opens a new line of inquiry with the potential to finally address the disparate 4-5:1 male to female ratio in autism [46, 47]. Most important yet, we will be able to do so using the test that researchers and clinicians use to detect autism, if we digitize it. Because of the wearable sensors’ revolution, we can easily digitize the ADOS today using off-the-shelf technology such as smart watches and web-cameras -an important advantage in remote testing and telemedicine approaches due to the pandemic. Under these new conditions, clinicians have limited visibility of the child-caregiver dyad at the other end of the screen. The connectivity metrics and the empirically derived Gamma-NSR are hidden to the naked eye, but visible to the digital data and to the analytics that we offer here. It will be important to scale up this result and validate it across thousands of dyads performing the ADOS test while capturing the dynamic dyad digitally. We here note that our methods extend to more than two people, so remote model of telemedicine involving the clinician at one end and the parent-child dyad at the other end, is also feasible.

We thank the reviewer for this comment. Since we had in the past reported such differences in the context of motor control, it was an oversight on our part to not ask so here. As it turns out this is arguably now the most important result of the paper! We note the result in the abstract too.

Reviewer 2 Report

This is a very interesting and novel paper on an important topic.  The merging of biosensor data with clinical observation appears to hold great promise in improving understanding of disorders of social perception and performance.

The paper overall provides sufficient detail and depth.  The rationale for the experiments is thorough.  The introduction would benefit from a clearer, concise statement of aim.  The aim seems implied but goals/hypotheses could be more explicit.

Within the Methods, please clarify if the two clinicians performing the ADOS-2 were trained to research reliability on the measure.  The ADOS can have reliability issues and research reliable training is gold standard to manage this limitation. 

How was the autism diagnosis for the autistic subjects determined and/or verified?  Where did these subjects come from (e.g., community referrals or other source)?

The primary criticism of the paper is that the authors do not acknowledge any limitations in their experiments, interpretation of findings, or statements of implications.  Their statements and interpretations appear generally justified, though clearly there are limitations and potential alternative explanations that should be enumerated to contextualize findings of this single site experiment on a small number of subjects.

Author Response

Reviewer 2

Comments and Suggestions for Authors

This is a very interesting and novel paper on an important topic.  The merging of biosensor data with clinical observation appears to hold great promise in improving understanding of disorders of social perception and performance.

We thank the reviewer for the very useful comments and address each one below

The paper overall provides sufficient detail and depth.  The rationale for the experiments is thorough.  The introduction would benefit from a clearer, concise statement of aim.  The aim seems implied but goals/hypotheses could be more explicit.

On line 123 we added the sentence We hypothesize that the digital ADOS will reveal new aspects of the shared space spanned by the cohesive dyadic micro-movements of the child and clinician.

Within the Methods, please clarify if the two clinicians performing the ADOS-2 were trained to research reliability on the measure.  The ADOS can have reliability issues and research reliable training is gold standard to manage this limitation.

On line 192 page 5 we specifically state this: Both clinicians had ADOS certification for research reliability.

How was the autism diagnosis for the autistic subjects determined and/or verified?  Where did these subjects come from (e.g., community referrals or other source)?

On page 4 line 177 we write: Autistic children were referred to our laboratory by local clinics and by the community, with help from the New Jersey Autism Center of Excellence (https://njace.us/), funded by the New Jersey Governor’s Council for the Medical Research and Treatments of Autism.

The primary criticism of the paper is that the authors do not acknowledge any limitations in their experiments, interpretation of findings, or statements of implications.  Their statements and interpretations appear generally justified, though clearly there are limitations and potential alternative explanations that should be enumerated to contextualize findings of this single site experiment on a small number of subjects.

On page 21 line 831 we have added a section to the Discussion where we address this weakness

4.1. Caveats and Limitations of the Proposed Hybrid Methods

Although the digital characterization of the ADOS test may hold promise to reveal new aspects of the ADOS interactions that may otherwise remain hidden to the naked eye, there are several caveats for the adoption and use of the present methods. The major limitation is the current lack of mechanisms that can scale up the results from this reduced cohort in the lab to thousands of participants in clinical settings. Future use of the analytics presented here can be extended to biorhythms registered with off-the-shelf wearables such as smart watches and web cameras outputting shareable digital data even across remote settings. To that end, our lab has developed a new concept “The Lab in a Box”. This remote kit for behavioral data acquisition and analyses offers continuity to the administration of behavioral tests such as the ADOS, in the times of social (physical) distancing. However, deploying such methods in large scale would require a concerted effort across multiple disciplines, business models and entrepreneurship beyond academic settings. This exceeds the scope of the paper, but it is perhaps something to keep in mind to translate the present results to clinical and home settings, offering the possibility to scale up the use of this technology to tens of thousands of users.

Another limitation of the present analyses is that other pipelines could have been used to analyze the data. This begs the question of whether the general results would have dramatically changed in that case. In earlier preliminary analyses of these data sets using different analytical pipelines, larger number of sensors in the interconnected grid, different bodily locations for the additional sensors, different kinematic parameters, and different children, we observed similar trends [24]. As such, we are confident that the trends would remain invariant to such nuances. However, we do acknowledge that the interpretation and inference are to be discussed and validated across the community, to provide a consensus on the clinical interpretability of digital data, in general. In other words, the characterization of the digital ADOS data that we provide here needs to be further validated across thousands of participants and clinicians. This is the case for any new scientific approach. In this sense, we invite the reader to reproduce the results using ours and other data sets provided in GitHub by our lab. Our analytics do not make theoretical assumptions about the likelihoods and probabilities to represent the data features. Our new methods rather empirically estimate these from the micro-fluctuations inherently present in the signals that we register from natural behaviors. In this sense, we remain confident that cross-sectional and longitudinal data capture will find trends like those found in this random draw of the population of autistics and neurotypical controls. This work extends an invitation to the clinical community as well as to the scientific community, to collaborate and take the steps toward combining different levels of inquiry in the study of general social behaviors beyond autism and other neurodevelopmental conditions.
